# Biochemical Markers for Neuroendocrine Tumors: Traditional Circulating Markers and Recent Development—A Comprehensive Review

**DOI:** 10.3390/diagnostics14121289

**Published:** 2024-06-18

**Authors:** Marianna Franchina, Federica Cavalcoli, Olga Falco, Marta La Milia, Alessandra Elvevi, Sara Massironi

**Affiliations:** 1Division of Gastroenterology, Fondazione IRCCS San Gerardo dei Tintori, 20900 Monza, Italy; 2Gastroenterology and Digestive Endoscopy Unit, Fondazione IRCCS Istituto Nazionale dei Tumori, 20133 Milan, Italy; 3Department of Medicine and Surgery, University of Milano-Bicocca, 20126 Milan, Italy

**Keywords:** neuroendocrine neoplasms, biochemical markers, molecular markers

## Abstract

Neuroendocrine neoplasms (NENs) are a heterogeneous group of neoplasms presenting unique challenges in diagnosis and management. Traditional markers such as chromogranin A (CgA), pancreatic polypeptide (PP), and neuron-specific enolase (NSE) have limitations in terms of specificity and sensitivity. Specific circulating markers such as serotonin and its metabolite 5-hydroxyindoleacetic acid (5-HIAA) and various gastrointestinal hormones such as gastrin, glucagon, somatostatin, and vasoactive intestinal peptide (VIP) have a role in identifying functional NENs. Recent advances in molecular and biochemical markers, also accounting for novel genomic and proteomic markers, have significantly improved the landscape for the diagnosis and monitoring of NENs. This review discusses these developments, focusing on both traditional markers such as CgA and NSE, as well as specific hormones like gastrin, insulin, somatostatin, glucagon, and VIP. Additionally, it covers emerging genomic and proteomic markers that are shaping current research. The clinical applicability of these markers is highlighted, and their role in improving diagnostic accuracy, predicting surgical outcomes, and monitoring response to treatment is demonstrated. The review also highlights the need for further research, including validation of these markers in larger studies, development of standardized assays, and integration with imaging techniques. The evolving field of biochemical markers holds promise for improving patient outcomes in the treatment of NENs, although challenges in standardization and validation remain.

## 1. Introduction

Neuroendocrine neoplasms (NENs) represent a heterogeneous group of tumors originating from neuroendocrine cells, which are distributed throughout various organs such as the gastrointestinal tract, pancreas, lungs, and thyroid gland. These tumors display a wide spectrum of clinical behaviors, ranging from indolent to highly aggressive forms [1,2,3]. NENs can be classified as functioning or non-functioning, depending on their ability to secrete hormones that lead to specific clinical syndromes [4,5,6]. The diagnosis and monitoring of NENs pose significant challenges due to their heterogenous presentations and the limitations of traditional diagnostic tools [6,7,8].

Traditional biochemical markers, including chromogranin A (CgA), neuron-specific enolase (NSE), and pancreatic polypeptide (PP), have been pivotal in the diagnosis and monitoring of NENs [9,10,11,12,13,14]. However, these markers often exhibit limited sensitivity and specificity, particularly in early-stage or non-functioning tumors. For instance, the sensitivity and specificity of CgA in detecting NENs can vary widely, affected by a range of factors such as concomitant medications and comorbid conditions, leading to potential false positives or negatives [15,16,17,18,19].

In contrast, recent advancements in molecular biomarkers, such as multigenomic blood assays, circulating tumor cells (CTCs), microRNAs (miRs), and long non-coding RNAs (lncRNAs), offer promising alternatives [20,21]. These new markers have shown improved sensitivity and specificity in the early detection, diagnosis, and monitoring of NENs. For example, the NETest^®^, a multianalyte liquid biopsy test, has demonstrated a sensitivity of over 90% and specificity close to 98%, significantly surpassing traditional markers [22,23]. Similarly, the analysis of circulating tumor DNA (ctDNA) and miRs provides insights into the genetic and epigenetic landscape of NENs, enabling not only accurate tumor detection but also the monitoring of disease progression and response to treatment [24,25].

This review aims to discuss the evolution of biochemical markers for NENs from traditional to contemporary molecular markers. We will explore their respective strengths and limitations, focusing on how the newer markers compare to their traditional counterparts in terms of sensitivity, specificity, and clinical utility. This overview sets the stage for a detailed discussion on the potential of these markers to revolutionize the clinical approach to managing neuroendocrine neoplasms.

## 2. Traditional General Biochemical Markers in NENs

The diagnosis and management of NENs have long been anchored in several traditional biochemical markers. Despite their fundamental role in clinical practice, these markers are not without limitations and often necessitate a multifaceted diagnostic approach. Understanding the strengths, limitations, and appropriate context for the use of these markers is critical to optimizing patient care.

### 2.1. Chromogranin A

Chromogranin A (CgA) stands as the most extensively used biomarker for NENs, attributed to its expression in neuroendocrine cells. CgA is an acidic glycoprotein found in the secretory granules of endocrine and neuroendocrine cells. It is co-released together with peptide hormones and biogenic amines and plays a central role in various biological processes [12]. CgA has been identified as a potential broad-spectrum biomarker for NENs as it correlates with tumor burden, progression, and metastasis. Its circulating level can indicate the presence and spread of NEN and provide valuable insights for diagnosis and management [13,26]. Despite its potential, the clinical utility of CgA is limited by issues affecting its specificity and sensitivity. These challenges stem from a variety of conditions that can lead to elevated CgA levels unrelated to NENs, resulting in both false-positive and false-negative results [27]. Since the secretion of CgA is ubiquitous [28], a variety of non-neoplastic processes leading to tissue damage and remodeling can cause an increase in the marker. These include a number of gastrointestinal diseases, such as chronic atrophic gastritis (CAG) [29], liver cirrhosis, chronic hepatitis [30,31], inflammatory bowel disease, and even irritable bowel syndrome [32]. Among cardiovascular diseases, increased CgA levels have been reported in hypertension, with higher levels detected in untreated patients [33]. This increase has also been reported in chronic heart failure, with higher levels detected in the fourth grade of the NHYA scale [34] and in acute coronary syndromes. Due to reduced clearance, an increase in this biomarker also occurs in renal and hepatic impairment [35]. In addition, hyperthyroidism and other endocrine and non-neuroendocrine disorders caused by increased sympathetic activity have also been reported to increase CgA levels [36]. Proton pump inhibitors (PPIs), other antacid drugs, and steroids are the main cause of an iatrogenic increase in CgA and may persist for up to two to three weeks after discontinuation of drug therapy [37]. Moreover, CgA is increased in patients with other malignancies such as prostate, breast, colorectal [38], gastric, pancreatic [39], and hepatocellular [30,31] cancers. The sensitivity of circulating CgA can be considered acceptable in functioning and advanced NENs. Indeed, CgA levels depend on the site of origin, with particularly high values for midgut carcinoids and pancreatic NENs, tumor burden, and secretory activity, with sensitivity and specificity ranging between 60–100% and 70–100% for different NENs, respectively [40]. Conversely, its sensitivity can be particularly low in localized, non-functioning diseases, posing a risk of false negatives. In addition, treatment with somatostatin analogs (SSAs) affects CgA levels, making it difficult to use as a reliable marker of tumor mass. Indeed, in patients treated with SSAs, there is no correlation between circulating CgA levels and tumor burdens. The reason for this is that SSAs are able not only to block the production and release of CgA but also have an antiproliferative effect [41,42]. A notable problem with the measurement of CgA is the variability between commercial assays, with sensitivities ranging from 67% to 93%. This inconsistency highlights the need for standardized approaches to improve the reliability of CgA assays.

These limitations highlight the need for testing CgA selectively (for instance, when a diagnosis has already been established) rather than as a screening tool to rule out the presence of neuroendocrine neoplasms (NENs).

The role of CgA in the management and follow-up of NEN patients has been reported in several studies [14]. CgA represents a reliable marker that is more useful for monitoring disease progression and response to treatment and for early detection of relapse after treatment rather than as part of the diagnosis [11,12,13,14]. Monitoring CgA levels over time can provide insight into treatment efficacy and signal tumor response or progression [11,12,13,14]. Indeed, changes in CgA levels may reflect modifications in tumor activity and burden even before imaging studies show any changes. This makes CgA a useful tool for tracking disease progression and adjusting treatment plans accordingly. Elevated baseline levels of CgA and their relative increase within the first year of observation have been identified as unfavorable predictors of overall survival, indicating the potential of CgA as a prognostic marker in patients with pancreatic and midgut neuroendocrine tumors undergoing peptide receptor radionuclide therapy with octreotide [43].

However, the interpretation of CgA levels during follow-up must consider potential confounders, such as concomitant medications and comorbid conditions that might affect CgA levels. Clinicians must correlate CgA trends with clinical findings and other diagnostic results for an accurate assessment of the patient’s status.

### 2.2. Pancreatic Polypeptide (PP)

Pancreatic polypeptide (PP) is secreted by PP cells in the pancreas and the gut mucosa, acting on G-protein coupled receptors. It was initially isolated from chickens in 1968 and humans in 1972 and found to be associated with some pancreatic neuroendocrine tumors in 1978 [44]. While PP can indicate the presence of pancreatic neuroendocrine tumors (panNENs), its levels can be falsely elevated in conditions like chronic kidney disease and gastrointestinal disorders, leading to false positives. This biomarker is used as a general biomarker of neuroendocrine differentiation, with a sensitivity of 54% in functioning tumors, 57% in non-functioning tumors, 63% in pancreatic tumors, and 53% in gastrointestinal (GI) tumors [45]. The specificity of PP as a diagnostic marker contrasts with a number of conditions that may lead to false elevations in PP levels, including gastrointestinal disorders such as diarrhea, laxative abuse, advanced age, inflammatory processes in the gut, and chronic kidney disease [46,47]. Despite these challenges, the combination of PP with CgA increases the diagnostic sensitivity for non-functional pancreatic NENs (panNENs) to almost 95%, highlighting the synergistic potential of biomarker combinations in neuroendocrine diagnostics [45,46,47]. As the release of PP is caused by the ingestion of meals, particularly those containing proteins, the clinical utility of PP extends beyond static measurement to dynamic testing, with the meal stimulation test serving as an important tool in the assessment of patients with multiple endocrine neoplasia type 1 (MEN1) and panNENs [41]. In addition, PP can also be used to predict disease progression as consistently reduced circulating PP levels indicate stable disease.

PP integration into the biomarker repertoire for NENs not only enhances diagnostic accuracy but also provides a functional approach to monitoring disease progression. On the other hand, it is important to note that PP assays are not readily available, significantly limiting PP utilization in clinical practice.

### 2.3. Neuron-Specific Enolase (NSE)

Neuron-specific enolase (NSE) stands as a relevant biomarker in the neuro-oncological and neuroendocrine diagnostic landscape. This enzyme, a neuron-specific isomer of the glycolytic enzyme 2-phospho-D-glycerate hydrolase, plays a critical role in the anaerobic conversion of glucose into energy and is a marker of neuroendocrine activity. Elevated serum levels of NSE are indicative of neuroendocrine malignancies, including small-cell lung carcinoma [48] and, albeit less frequently, Merkel cell tumors [49]. NSE expression has been consistently linked to poor tumor differentiation [49]. However, the specificity of NSE is challenged by its expression in a wide range of neuroendocrine and non-neuroendocrine tissues, which may contribute to false positives. Moreover, the study called RADIANT-1 shows that early CgA or NSE response can be useful as a potential predictor of treatment outcomes for patients with advanced pNET treated with Everolimus. In addition, these early responders to biomarkers were also more likely to show better responses to antitumor therapy [50].

## 3. Specific Circulating Hormones

A variety of hormonal markers are employed in the diagnosis and monitoring of functioning NENs, which produce symptoms related to hormone hypersecretion. These include insulin, gastrin, vasoactive intestinal peptide (VIP), glucagon, and somatostatin, among others. The measurement of these hormones plays a critical role in diagnosing specific syndromes associated with NENs, such as insulinoma, Zollinger-Ellison syndrome, VIPoma, and glucagonoma, providing a direct link between clinical symptoms and tumor activity (Table 1).

### 3.1. Gastrin

Gastrin, secreted by gastric G cells, primarily stimulates acid secretion and plays a significant role in digestive processes and gastric mucosal growth [51]. Gastrinoma is the second most common functioning NEN. It is predominantly located within the “gastrinoma triangle” (duodenum, pancreas, and extra-hepatic biliary system). Its pathological overproduction of gastrin leads to Zollinger–Ellison syndrome (ZES), characterized by excessive gastric acid secretion, gastroesophageal reflux disease, recurrent peptic ulcers, and chronic diarrhea [52,53]. Diagnosis hinges on elevated fasting serum gastrin levels (>1000 pg/mL) and gastric pH below 2, with the secretin and glucagon provocative tests serving as confirmatory diagnostics under specific conditions [52].

The measurement of gastrin levels plays a pivotal role in diagnosing gastrinomas and ZES. However, elevated gastrin levels are not exclusively indicative of ZES, as they can also be caused by various non-neoplastic conditions, including chronic use of PPIs, atrophic gastritis, and renal failure. This overlap greatly complicates the diagnostic process and requires a deep understanding of the sensitivity and specificity of gastrin in the context of ZES [53].

In particular, while PPIs are effective in treating symptoms of acid hypersecretion, they pose a major challenge in the diagnosis of ZES, as their use effectively masks the clinical and biochemical signature of ZES. On the other hand, discontinuation of PPIs in patients with suspected ZES is fraught with difficulty due to the worsening of symptoms and increased risk of peptic complications such as bleeding and perforations [52,53]. Thus, this clinical scenario requires careful management to avoid misdiagnosis. Recently, current guidelines have been updated to overcome the need to discontinue PPIs for ZES diagnosis [54]. Indeed, it is now recognized that the secretin stimulation test, a crucial diagnostic tool for ZES, can be performed without discontinuing PPI therapy [55]. This modification acknowledges that the test evaluates the variation (delta) in gastrin levels rather than the absolute values. This approach enables more refined diagnostic criteria in the presence of concurrent PPI use [56]. Secretin provocative test can be used in patients with suspected ZES when gastric pH is <2 but gastrin levels < ×10 upper limit of normal. The test consists of the administration of secretin (2 U/kg body weight) by intravenous bolus; fasting serum gastrin is measured at baseline (15 and 1 min before the test) and then 2, 5, 10, 15, 20, and 30 min after secretin administration. An increase of ≥120 pg/mL at any time during the test confirms the diagnosis [7,57]. However, due to difficulties in sourcing secretin and potential risks associated with the test, its usage has declined in recent times [5].

The glucagon provocative test is performed by administering glucagon (20 μg/kg) intravenously, followed by 20 μg/kg for the next 30 min, and plasma gastrin levels are measured 3 and 1 min before and 3, 5, 7, 10, 15, 20, and 30 min after the administration of glucagon. Diagnosis of gastrinoma should be made when plasma gastrin levels peak within 10 min after glucagon administration, with an increase of greater than 200 pg/mL and greater than 35% of the basal value [58].

Localization of the primary tumor and its metastases is the next diagnostic step when gastrinoma-associated ZES is either suspected or biochemically confirmed. Endoscopic ultrasound has showcased sensitivity as high as 83% for pancreatic gastrinomas and is considered the primary modality in such cases, although its tumor detection rates are substantially lower in duodenal lesions.

High-resolution imaging, especially Gallium-68 DOTATOC PET, is key for localizing gastrinomas after biochemical confirmation [51,52,53,58,59], enhancing traditional imaging techniques, and showcasing high sensitivity and specificity [52,53]. 

### 3.2. Insulin

Insulin is a peptide hormone with anabolic properties produced by the β cells of the islets of Langerhans within the pancreas. The active form of insulin is synthesized from the proinsulin precursor molecule and consists of two peptide chains: A-chain and B-chain. Insulin plays a key role in energy balance and glucose metabolism, mainly reducing blood glucose levels by increasing glycogen synthesis and promoting the storage of glucose in the liver and muscle cells.

Insulinomas arise almost exclusively from the pancreas and are the most common pancreatic functioning NEN. They are classically characterized by the so-called Whipple’s triad, a specific diagnostic hallmark consisting of 3 main features: low plasma glucose concentration (<45 mg/dL); clinical signs/symptoms consistent with hypoglycemia; and resolution of them when the plasma glucose concentration increases [60,61].

Clinical suspicion of an insulinoma should be confirmed using the 72-h fasting test, which is the gold standard for diagnosing insulinoma, given its sensitivity of 100%. The test requires hospitalization and placement of intravenous access. During the test, a blood sample is drawn every 6 h for glucose and contextual insulin level determination until blood glucose levels drop to ≤45 mg/dL and insulin levels simultaneously reach ≥36 pmol/L [5,62]. Of note, insulin concentrations may be within the reference range; however, insulin results are inappropriately high for the blood glucose level. Furthermore, tumors that produce insulin also release C-peptide and proinsulin, which are detectable in blood samples. These markers can help distinguish between tumor secretion and an insulin overdose.

In the case of a nonconclusive test, it is possible to perform a glucagon stimulation test. After administration of glucagon (1 mg), an increase in serum glucose indicates adequate stores of glycogen and consequently confirms the presence of an insulinoma [5,7,62].

### 3.3. Serotonin and 5-Hydroxyindoleacetic Acid (5-HIAA)

Urinary 5-HIAA is the urinary metabolite of serotonin and represents a cornerstone in diagnosing and monitoring the treatment of neuroendocrine tumors, particularly those located in the small bowel and associated with carcinoid syndrome. It reflects the tumor’s serotonin production capacity and is used for a comprehensive assessment of these neoplasms [9,63,64]. Serotonin (5-hydroxytryptamine, 5-HT) is a derivative of the amino acid tryptophan and is produced mainly in the nervous system, platelets, and enterochromaffin cells within the digestive system (mostly in the small bowel). Serotonin is also involved in different biological functions, including vasoconstriction, neurotransmission, regulation of sleep, appetite, and gastrointestinal motility, exerting a prokinetic effect on the gastrointestinal tract; it stimulates fluid secretion and causes nausea and vomiting by stimulating smooth muscle and sensory nerves in the stomach [65]. Serotonin secretion undergoes fluctuations over time and depends on the individual. Therefore, it is difficult to monitor. In consideration of the low specificity, evaluation of circulating and urinary serotonin is not recommended [66]. While urinary 5-HIAA remains a valuable marker, its measurement presents challenges, including the inconvenience of 24-h urine collection and dietary restrictions to avoid false positives. A three-day diet free of food rich in tryptophan/serotonin (i.e., tomatoes, plums, pineapples, bananas, eggplants, avocados, and walnuts) is advised to avoid false elevations in urinary 5-HIAA. In addition, drugs such as diazepam and phenobarbital may lead to a false-positive result [67]. Moreover, false positives can be observed due to malabsorption and celiac disease, while renal insufficiency can, in turn, lead to false negatives [68]. False-negative results have also been reported in patients on medications such as levodopa, methyldopa, acetylsalicylic acid, adrenocorticotrophic hormone (ACTH), and phenothiazines. 5-HIAA showed a high specificity of ∼100% but low sensitivity (~35%) due to diverse expression in individuals [68].

Recent advancements in the measurement of 5-HIAA in serum or plasma offer a promising alternative, potentially simplifying the monitoring process and improving patient compliance [63].

Moreover, a prognostic value of 5-HIAA in patients with carcinoid syndrome has been proposed. In 2016, Zandee WT et al. found a significant correlation between high urinary 5-HIAA levels and survival outcomes, with higher levels predicting shorter progression-free survival (PFS) and overall survival (OS) at univariate analysis, although the result was not confirmed at multivariate analysis [69]. In 2007, Formica et al. showed that 5-HIAA urine concentration shows a positive correlation with the severity of carcinoid syndrome [70] and also the response to treatment, particularly with SSAs and targeted therapies [70,71,72]. Therefore, it is used both for diagnosis as well as a type II marker of disease course [73].

In conclusion, 5-HIAA represents a useful marker in the diagnosis and follow-up of serotonin-producing tumors mostly located in the small bowel and respiratory tract and a minority located in the pancreas [74].

### 3.4. Glucagon

Glucagon is a peptide hormone composed of 29 amino acids, primarily produced by the α cells of the pancreatic islets and L cells in the intestinal mucosa. It plays a crucial role in glucose homeostasis by stimulating gluconeogenesis and glycogenolysis [75]. Glucagonoma is a rare type of neuroendocrine tumor that predominantly occurs in the pancreas’s islets of Langerhans and typically arises from the tail or the body of the pancreas due to the high prevalence of alpha cells in this area and causes the unregulated secretion of glucagon, elevating blood glucose levels by enhancing gluconeogenesis and lipolysis. It has an annual incidence ranging from 0.01 to 0.1 per 100,000 [5,76]. A distinctive necrolytic migratory erythematous rash, angular stomatitis, painful glossitis, and normochromic normocytic anemia characterize the glucagonoma syndrome. It is also characterized by mild diabetes mellitus, weight loss, a predisposition to thrombosis, and neuropsychiatric disturbances. This syndrome is exceedingly rare, with an estimated incidence of one in 20 million people globally [77].

The diagnosis of glucagonoma syndrome relies on identifying elevated plasma glucagon concentrations, typically above 500 pg/mL (normal value < 150 pg/mL) [78], in the absence of other causative factors like renal failure or severe stress. Immunocytochemistry employing glucagon antibodies aids in pinpointing and characterizing the pancreatic alpha-cell tumor.

### 3.5. Somatostatin

Somatostatin is a peptide hormone produced by the delta cells of the pancreas, the gastric antral and oxyntic D cells, and the APUD (Amine Precursor Uptake and Decarboxylation) cells [79]. Somatostatin inhibits numerous endocrine and exocrine secretory functions. Almost all gut hormones are inhibited by somatostatin, including insulin, glucagon, gastrin, secretin, and gastric inhibitory polypeptide (GIP). In the nervous system, somatostatin acts as a neurotransmitter or neuromodulator.

Somatostatinomas represent less than 1% of functioning gastrointestinal NENs, and their estimated incidence is about 1 in 40 million individuals per year [79]. Somatostatinoma, originating predominantly in the pancreas or duodenum, presents a unique challenge due to their rarity and the broad spectrum of symptoms associated with them, such as diabetes, steatorrhea, and cholelithiasis. Somatostatinoma can be sporadic or may occur in association with familial syndromes such as MEN1 (40% to 50% of cases), neurofibromatosis type 1, and Von Hippel-Lindau syndrome. Elevated somatostatin levels in the blood reflect the secretory activity of the tumor, and the ability to detect and quantify circulating somatostatin levels offers a good tool for the diagnosis and management of somatostatinomas. However, the presence of somatostatin-producing cells in various tissues and the hormone’s role in normal physiological processes can affect the specificity of circulating somatostatin as a sole marker for somatostatinomas. Therefore, the diagnosis of somatostatinoma requires the combination of typical clinical signs and symptoms with measuring the fasting plasma somatostatin hormone concentration, which should be at least three times over the upper reference value (>25 pmol/L or >60 pg/mL) [79].

### 3.6. Vasoactive Intestinal Peptide (VIP)

The vasoactive intestinal polypeptide (VIP) is a neurotransmitter synthesized in the central nervous system and neurons in the intestine, lungs, adrenals, pancreas, and liver. On the digestive system, VIP has several effects: vasodilatation; smooth muscle contraction; stimulation of water and electrolyte secretion from the intestinal tract; inhibition of gastric acid secretion; and increase in blood flow in the GI tract. VIP-secreting tumors, VIPoma, are rare tumors occurring both in children and adults, with an incidence ranging from 0.05% to 2.0% [80]. VIP hypersecretion causes severe watery secretory diarrhea, which can result in hypokalemia and metabolic acidosis (VIPoma syndrome or “Verner-Morrison syndrome”). Hypochlorhydria, stimulation of glycogenolysis, and hypercalcemia can also be found in VIPoma patients. The majority of VIPomas are located in the pancreas (75%), and patients can present with VIP-producing neuroblastoma, ganglioneuroblastoma, ganglioneuroma, pheochromocytoma, and paraganglioma, or neoplasms of the retroperitoneum and mediastinum [81]. Assessment of VIP levels is often sufficient to diagnose a patient with VIPoma since elevated VIP levels (>60 pmol/L) achieve a specificity of 100% [68].

### 3.7. Adrenocorticotropic Hormone (ACTH)

ACTH is an aminoacidic hormone secreted from the anterior pituitary gland in response to the hormone corticotropin-releasing hormone (CRH) released from the hypothalamus. This hormone targets the cortical area of the adrenal gland and stimulates the secretion of glucocorticoids. It is part of the hypothalamic–pituitary–adrenal axis and regulates metabolism, rest, and drowsiness. Chronic exposure to excess glucocorticoids results in diverse manifestations of Cushing’s syndrome, such as increased body weight, muscle weakness, hypertension, hyperglycemia, hypokalemia, infections, bruising, osteoporosis, and psychiatric disorders. Endogenous Cushing’s syndrome is classified into ACTH-dependent forms (about 80%) and ACTH-independent (about 20%) forms (Table 1) [82]. ACTH ectopic secretion accounts for 10% to 20% of all cases of Cushing syndrome. The most common sources of ectopic ACTH secretion (ACTH-omas) are usually small-cell lung cancer, bronchial carcinoid, medullary thyroid cancer, and pheochromocytoma [82,83].

### 3.8. Corticotropin-Releasing Hormone (CRH) and Growth-Hormone-Releasing-Hormone (GHRH)

CRH is a 41 aminoacidic hormone produced by neuroendocrine cells in the paraventricular nucleus of the hypothalamus. It stimulates ACTH secretion in response to stress [84].

CRH-secreting tumors are rare. The tumors associated with ectopic CRH secretion are pheochromocytoma, medullary thyroid carcinoma, and prostate cancer [85]. The clinical presentations of ectopic CRH syndrome might be similar to that of ectopic ACTH syndrome, so the differential diagnosis of these two diseases is important [85].

GHRH is a 44 amino acid hormone produced in the arcuate nucleus of the hypothalamus and regulates the release of growth hormone (GH). Excessive secretion of GHRH by a tumor leads to somatotroph cell hyperstimulation, increased GH secretion, and acromegaly. Several hypothalamic tumors, including hamartomas, gliomas, and gangliocytomas, can produce GHRH. However, in this case, peripheral levels of GHRH are typically not elevated because the hormone secreted into the hypophyseal portal system does not reach the systemic circulation. On the other hand, excessive ectopic peripheral production of GHRH usually causes elevation of peripheral GHRH levels. Ectopic peripheral production of GHRH has been reported in several tumors, including pulmonary NEN and small-cell lung cancers (54%), pancreatic NEN (30%), small-intestine NEN (7%), adrenal adenomas, and pheochromocytomas. The distinction of pituitary vs. extra-pituitary acromegaly is extremely important in planning effective management. Regardless of the cause, GH and IGF-1 are invariably elevated, and GH levels fail to be suppressed (<1 microg/L) after an oral glucose load in all forms of acromegaly. Plasma GHRH levels are usually elevated in patients with peripheral GHRH-secreting tumors and are normal or low in patients with pituitary acromegaly. Unique and unexpected clinical features in an acromegalic patient, including respiratory wheezing or dyspnea, facial flushing, peptic ulcers, or renal stones, are sometimes helpful in alerting the physician to a nonpituitary endocrine tumor diagnosis [86].

### 3.9. Calcitonin

Calcitonin is a hormone consisting of a polypeptide of 32 amino acids, which is produced by the parafollicular cells of the thyroid (also known as C cells) and by K cells of the respiratory epithelium. The main function of calcitonin is lowering blood calcium concentration, in contrast with the parathyroid hormone (PTH). The most common cause of an increased calcitonin serum level is medullary thyroid cancer, which frequently arises as part of MEN type 2 (MEN2) syndrome. Nevertheless, calcitonin may be raised in other solid neoplasms, including pancreatic and pulmonary NENs, pheochromocytomas, neuromas, breast, prostate, and colorectal carcinomas. Calcitonin values >100 pg/mL are strongly suspicious of malignancy, whereas, in patients with moderately elevated values (10–100 pg/mL), a stimulation test may be applied to improve diagnostic accuracy. Patients with medullary thyroid cancers show stimulated calcitonin values at least three to four times higher than the basal values, whereas calcitonin-secreting NENs can be distinguished from a C-cell disease by the absence of or lower response to stimulation [87].

These specific circulating markers provide invaluable insights into the physiological disruptions caused by functioning NENs, facilitating targeted diagnostic, therapeutic, and monitoring strategies. Their application underscores the tailored approach needed in managing the diverse spectrum of neuroendocrine tumors.

## 4. Advances in Molecular Biochemical Markers

Due to the limitations of classical biomarkers, in recent years, the search for new biomarkers has focused on molecular methods that measure multi-analytes through statistical algorithms. These analytes are genomic and proteomic biomarkers that can be detected in peripheral blood according to the method of liquid biopsy [21].

The landscape of diagnosis, prognosis, and treatment of NENs has recently been significantly revolutionized by the identification and application of molecular markers. These markers have provided insights into the genetic and molecular pathways involved in the development and progression of NETs and offer new opportunities for targeted therapy and personalized medicine.

### 4.1. Genomic Alterations in NENs: Mutations, Gene Expressions and Epigenetic Changes

The discovery of genomic alterations has greatly expanded our understanding of NENs. These alterations, including mutations, gene expressions, and epigenetic changes, play a crucial role in the behavior of NENs. These new insights have been crucial in identifying potential therapeutic targets and prognostic indicators.

Molecular markers developed from these studies include the neuroendocrine neoplasms test (NETest^®^), circulating tumor cells (CTCs), circulating tumor DNA (ctDNA), and microRNAs (MiRs) (Table 2).

#### 4.1.1. The NETest^®^

The NETest^®^ is an advanced multianalyte liquid biopsy that quantifies gene expression from NENs in blood samples, providing real-time insights into the biological activity and genetic signature of the disease. It operates by analyzing 51 specific genes known to be associated with tumor proliferation, signaling, secretion, and overall neoplastic behavior, selected through rigorous microarray technology research. These genes, often overexpressed in various NEN types, are isolated as messenger-RNAs from a patient’s peripheral blood, converted into complementary-DNA, and then amplified using polymerase chain reaction techniques [21]. Utilizing sophisticated algorithms, the NETest^®^ quantifies these genes and evaluates the expression levels of precise gene clusters. The results are then transformed into a percentage known as the NETest^®^ score, which indicates the presence and activity level of the disease. This score is highly standardized and reproducible, unaffected by patient-specific factors such as age, gender, ethnicity, fasting state, or medications [21,88]. It correlates better than traditional biomarkers with imaging, tumor grade, and ki67 index, and its accuracy makes it a potent tool for diagnosis and prognosis [89].

Clinically, the NETest^®^ has demonstrated superior correlation with imaging techniques, tumor grade, and ki67 index compared to traditional biomarkers. Its diagnostic accuracy is significantly higher—up to ten times more accurate than CgA—with a sensitivity of 93.2%, a specificity of 98.4%, and an overall accuracy of 95.6%. These capabilities are confirmed by its strong correlation with both traditional and functional imaging in over 90% of cases [20,23,89,90]. This high diagnostic accuracy has been confirmed in a meta-analysis and is independent of tumor stage or grade [89,91,92].

The prognostic value of the NETest^®^ is equally promising; it effectively correlates with tumor burden and disease activity. Higher scores indicate more aggressive disease states, with scores between 21 and 40% suggesting stable disease and those between 41 and 100% indicating progressive disease [93,94,95]. This prognostic accuracy has been validated across different NEN types [91,95,96], showing that the NETest^®^ is effective in monitoring disease progression and predicting responses to treatments such as surgery, embolization, SSAs, and peptide receptor radionuclide therapy (PRRT) [93,97,98]. Notably, reductions in the NETest^®^ score post-treatment suggest successful therapeutic interventions, and an increase in score typically signals minimal residual disease or recurrence with high accuracy [23,94,99].

Additionally, the NETest^®^ can detect residual disease, recurrence, and metastasis approximately six months earlier than radiological and traditional biomarkers, providing a critical window for earlier intervention [98,100]. In treatment strategy, NETest^®^ scores below 40% and above 80% have shown correlations with non-responsiveness and responsiveness to SSAs, respectively, while scores over 40% predict a response to PRRT [97,101,102].

Ultimately, the NETest^®^ offers a powerful tool for non-invasively monitoring and managing neuroendocrine tumors, facilitating early detection and tailored treatment strategies, potentially reducing the reliance on more invasive diagnostic methods like radiological imaging.

#### 4.1.2. Circulating Tumor Cells (CTCs)

CTCs are neoplastic cells found in the bloodstream, providing a minimally invasive avenue for detecting and characterizing NENs. Despite their potential, the clinical utility of CTCs in terms of diagnostic accuracy and prognostic value remains somewhat uncertain due to inconsistent detection rates and mixed outcomes across studies.

Approximately 50% of patients with NENs have detectable CTCs, with their presence and concentration often linked to higher tumor grades and increased tumor burden [103,104]. The detection of CTCs in these patients has been associated with shorter progression-free survival (PFS) and poorer overall survival (OS), suggesting a potential prognostic role [103,104,105]. However, their reliability as a diagnostic tool is compromised by the fact that they are only identified in half of the cases with confirmed metastatic NENs, leading to questions about their overall diagnostic accuracy and prognostic value [103].

Indeed, research indicates varying results in relation to the presence of CTCs. Some studies have shown that a high count of CTCs at diagnosis is associated with shorter OS, while lower numbers and a decrease in numbers after treatment generally indicate a favorable prognosis [106,107]. In contrast, some results, such as those from a study examining the effect of lanreotide, showed no significant differences in response rates or PFS between patients with and without detectable CTCs, highlighting the complex role of CTCs in the treatment of NENs [108].

The predictive ability of CTCs has also been explored, particularly through the detection of specific mutations and copy number changes corresponding to those in the primary tumor [109]. Notably, recent studies have identified therapeutic targets on CTCs, such as somatostatin receptor types 2 and 5, suggesting a potential role in predicting response to treatment [110]. However, the heterogeneity within CTC populations and the potential for subclonal variation pose challenges to fully delineating the characteristics of the primary tumor. In summary, while CTCs represent a promising tool for non-invasive monitoring of NENs, their exact role in clinical practice needs to be further clarified through ongoing research and validation, given their current limitations in terms of diagnostic accuracy and prognostic reliability.

#### 4.1.3. Circulating Tumor DNA (ctDNAs)

ctDNA comprises short nucleic acid fragments with a length of about 150 base pairs that are released into the bloodstream by apoptotic, necrotic, and autophagic cell processes. Detectable in body fluids as free DNA, protein-bound, or extracellular vesicles, ctDNA provides a non-invasive insight into the genetic and epigenetic composition of NENs [111]. Its rapid turnover facilitates real-time monitoring of tumor dynamics and makes ctDNA a valuable biomarker for the assessment of disease progression and response to therapy. The presence of ctDNA in patients with NENs is indicative of several tumor characteristics, including primary tumor location, metastatic status, and higher tumor grades, and is associated with poorer prognosis, including lower OS and shorter PFS [112,113,114]. Elevated ctDNA levels generally correlate with advanced disease stages, higher tumor burden, and increased proliferation indices [113,115]. Quantitative analysis of ctDNA can aid in the assessment of tumor volume and could potentially support therapeutic decisions by providing a non-invasive method to monitor treatment efficacy and disease progression [116,117,118,119]. In addition, qualitative analysis of ctDNA, including the identification of specific mutations and hypomethylation patterns, helps to predict disease prognosis and response to therapies [24]. Despite these advantages, the detection of ctDNA in NENs is associated with challenges due to the typically slow growth of these tumors, which can lead to false-negative results, representing a significant limitation of diagnostic utility [120,121].

Overall, the application of ctDNA in the treatment of NENs underscores its potential as a dynamic biomarker that reflects the genetic landscape of the disease and supports personalized medicine by providing timely insights into tumor behavior and response to treatment. Further validation and integration of ctDNA analysis into clinical practice is needed to fully realize its potential to improve NEN management.

#### 4.1.4. MicroRNAs (MiRs)

MiRs are small, non-coding RNAs that play an important role in the post-transcriptional regulation of gene expression and influence various cellular processes, including carcinogenesis [122,123]. These molecules serve as promising diagnostic and prognostic biomarkers for NENs and can be effectively identified using liquid biopsy techniques. Different miR expression profiles are associated with specific NEN types and sites, which increases their potential for targeted diagnostic and prognostic applications. For example, specific miRs such as miR-125a, miR-99a, and miR-99b are associated with pancreatic NENs, miR-204 with insulinomas, and miR-375 with gastroenteric NENs. Furthermore, miR-1290 is crucial for distinguishing pancreatic cancer from benign pancreatic diseases and other NENs [124,125,126,127]. Overexpressed miR-210 is associated with a high Ki67 proliferation index and the presence of liver metastases, indicating its potential as a prognostic marker. The relationship between miR-29b and CgA levels is further evidence of the interconnected role of miRs in the biology of NENs [126,128,129,130].

However, the clinical application of miRs is contingent on further validation and the development of standardized detection methods. To date, numerous technological approaches have been used for the detection of miRs, highlighting the need for precise and reliable tests based on mathematical algorithms to improve their clinical viability. Future studies are needed to fully establish their role in tumor biology and their potential clinical applications [115,128,131].

The study of extracellular vesicle-derived miRs offers new insights into the stability and disease relevance of these biomarkers. These vesicle-entrapped miRs, which are protected from enzymatic degradation, offer a promising avenue for non-invasive, stable, and specific diagnostic tools for the treatment of NENs [132].

### 4.2. Proteomic Biomarkers

Proteomic studies using mass spectrometry have identified a number of potential biomarkers in NENs, including specific cytokines and transmembrane receptors that can be determined in peripheral blood. In particular, vascular endothelial growth factor (VEGF) and its receptors (VEGFR-1, VEGFR-2, VEGFR-3), interleukin-8 (IL-8), and stromal cell-derived factor-1α (SDF-1α) are key angiogenic factors that promote vascularization and the development of NENs [133]. Elevated baseline levels of VEGFR-2 and SDF-1α are associated with poorer prognosis, while high levels of IL-8 are associated with better response to treatment, highlighting the complexity and importance of these biomarkers for patient management [134,135,136,137].

Other proteins have been isolated as specific diagnostic biomarkers for certain types of NENs. For example, the upregulation of actin-related protein 3 (ACTR3), CD163, and leukocyte cell-derived chemotaxin 2 (LECT2), in addition to the downregulation of complement C1q-A chain (C1QA) and cartilage oligomeric matrix protein (COMP), have been associated with pancreatic NENs [138]. These results suggest that these proteins could serve not only as diagnostic markers but also as therapeutic targets. However, the clinical utility of these biomarkers still needs to be validated by extensive studies to definitively clarify their role in the diagnosis and treatment of NENs. However, more studies are still required in order to assess their real utility in clinical practice.

Ongoing advances in proteomics continue to unravel the complex protein profiles of NENs, contributing to a deeper understanding of their pathophysiology and aiding in the development of targeted therapies. As research progresses, these proteomic biomarkers are expected to improve the precision of NEN diagnosis and therapy and pave the way for more personalized treatment strategies for patients.

## 5. Clinical Applicability and Outcomes

The possibility of detecting genomic and proteomic biomarkers through a blood test opens interesting scenarios in the diagnosis and management of NENs.

Indeed, liquid biopsy and protein measurement are non-invasive and can be easily performed in all patients, even in cases where a standard biopsy may not be possible or when the primary tumor has not been identified.

Moreover, they can be repeated to monitor tumor changes in real time and to predict and monitor response to treatment.

The identification of specific molecules can significantly influence therapeutic choices, and monitoring these changes can help in identifying mechanisms of resistance to treatments.

By using mathematical algorithms and characterizing their molecular features, new biomarkers provide a more comprehensive depiction of the disease state. Furthermore, they are not influenced by gender, age, food, and medications, which typically impact the accuracy of traditional markers [64,139].

The strongest evidence of the usefulness of the new markers comes from NETest^®^, which stands out as the only one that is standardized and reproducible.

The NETest^®^ has been shown to be useful in diagnosis, prognosis, assessing treatment responses including surgical radicality, and detecting disease progression and recurrence [109,120]. While similar evidence exists for other new biomarkers, further research is required to determine their definitive roles.

The main limitations of these new methods are the lack of standardization, limited availability, and high costs. In fact, they are not widely used in clinical practice due to the specialized laboratory technology and expertise required and their higher expense compared to traditional markers. However, they have the potential to offset costs by reducing the need for other diagnostic tests and therapies [139,140].

## 6. Challenges and Future Directions

The management and diagnosis of NENs are evolving, facing major challenges and opening promising new directions for the future. Despite advances in the field of biomarkers, numerous difficulties remain, including the variability in marker expression, the need to improve sensitivity and specificity, and the integration of new diagnostic modalities into clinical practice.

A major hurdle is the heterogeneity of NEN, which necessitates a personalized approach to diagnosis and treatment. The variability in tumor behavior and response to therapy underscores the need for more precise markers. Future research should focus on identifying new markers that are able to predict response to treatment and monitor disease progression more accurately.

Given the limitations of biomarkers, generic circulating markers currently play a limited diagnostic role. Due to numerous false results, they do not allow early or accurate diagnosis; on the contrary, their inappropriate use often leads to unnecessary diagnostic expenses. Their role could be useful only in follow-up as an early indicator of recurrence, perhaps to decide whether imaging studies should be accelerated or delayed. In this sense, strategic studies should be conducted to evaluate their actual benefit. Conversely, specific markers are useful in diagnosing functioning forms of NENs, which often require a different approach or treatment than non-functioning forms. Therefore, their use should be encouraged when there is a diagnosis of NEN and clinical signs consistent with a functioning neoplasia syndrome (ZES, Verner Morrison, VIPoma, etc.) (Figure 1).

A promising area is the use of liquid biopsy 6techniques for real-time disease monitoring. The potential to detect ctDNA, miRs, and CTCs provides a non-invasive way to understand tumor dynamics, evaluate treatment efficacy, and detect relapse early. However, the standardization of these methods and the interpretation of their results in the context of NENs will be critical. In addition, the integration of genomic and proteomic data promises to reveal new therapeutic targets and facilitate the development of targeted therapies. This precision medicine approach could revolutionize the treatment of NENs by making it more effective and reducing side effects.

Collaboration between clinicians, researchers, and patients is essential to overcome these challenges. Multidisciplinary efforts are needed to translate research findings into clinical practice to improve outcomes for patients with NENs.

## 7. Conclusions

In summary, there are several challenges in the field of circulating markers for NENs. The evolving landscape of biochemical markers for NENs offers new opportunities for the accurate diagnosis, prognosis, and management of these tumors (Figure 1). Traditional markers, despite their utility, have limitations in specificity and sensitivity, driving the need for more refined molecular biomarkers. Recent advances in genomic and proteomic technologies have led to the development of more precise biomarkers that enhance our understanding of NENs and enable personalized treatment strategies. However, the adoption of these new markers in clinical practice faces hurdles due to standardization challenges, cost, and availability. Future efforts should focus on validating these innovative markers in larger cohorts and standardizing testing procedures.

## Figures and Tables

**Figure 1 diagnostics-14-01289-f001:**
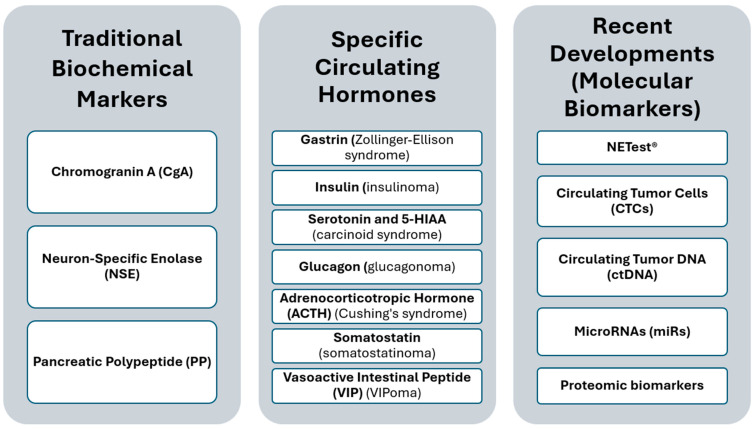
Comprehensive Overview of Biochemical Markers in Neuroendocrine Tumor.

**Table 1 diagnostics-14-01289-t001:** This table highlights the types of specific markers and their clinical utility in the diagnosis and treatment of various neuroendocrine neoplasms and illustrates their significance in linking clinical symptoms to tumor activity and guiding treatment strategies.

Marker	Associated Syndrome	Utility
Gastrin	Zollinger-Ellison syndrome (ZES)	Diagnosis and monitoring of gastrinomas, crucial for identifying ZES.
Insulin	Insulinoma	Confirming insulinoma presence, particularly through the 72-h fasting test.
Serotonin and 5-HIAA	Carcinoid syndrome	Key in diagnosing and managing carcinoid syndrome, reflecting serotonin production.
Glucagon	Glucagonoma	Identifying glucagonoma through elevated plasma glucagon levels.
Somatostatin	Somatostatinoma	Diagnosing somatostatinomas, monitoring elevated somatostatin levels.
VIP	VIPoma (Verner-Morrison syndrome)	Essential for diagnosing VIPoma, indicated by severe watery diarrhea and elevated VIP levels.
ACTH	Cushing’s syndrome	Diagnosing ACTH-dependent forms of Cushing’s syndrome.
CRH	Ectopic CRH syndrome	Important in differentiating ectopic CRH syndrome from ectopic ACTH syndrome.
GHRH	Acromegaly	Useful in diagnosing acromegaly caused by GHRH-secreting tumors.
Calcitonin	Medullary thyroid cancer (MTC)	Suggestive of MTC, particularly in patients with elevated calcitonin levels.

**Table 2 diagnostics-14-01289-t002:** This table highlights the role of molecular markers in the diagnosis, prognosis, and treatment of NENs, providing new opportunities for personalized medicine.

Marker	Description	Clinical Utility
NETest^®^	A multianalyte liquid biopsy measuring 51 different genes’ expression related to NEN activity.	High diagnostic accuracy, predictive of treatment response, and disease monitoring.
CTCs	Neoplastic cells in the bloodstream, indicative of tumor grades and burden.	Potential in tumor detection and prognosis, though with variability in detection and outcomes.
ctDNAs	Short nucleic fragments in body fluids, providing insights into the genetic composition of NENs.	Allows real-time monitoring of tumor development, with levels correlating with disease stages.
MiRs	Small non-coding RNAs regulating gene expression, linked to specific NEN types.	Offers diagnostic and prognostic value, with specific miRs associated with different NEN types.
Proteomic biomarkers	Identified via mass spectrometry, including cytokines and receptors like VEGF and its receptors.	Potential diagnostic markers and therapeutic targets, though further studies are needed.

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
