# Peer review of "Biochemical Markers for Neuroendocrine Tumors: Traditional Circulating Markers and Recent Development—A Comprehensive Review"

_diagnostics, 2024, doi:10.3390/diagnostics14121289_

Round 1

Reviewer 1 Report

Comments and Suggestions for Authors

The paper reviews the current data on biochemical markers for neuroendocrine tumors.

Neuroendocrine neoplasms represent a constantly changing field, whose dynamics has significant impact on patient’s management.

The paper is well-structured and begins with traditional biomarkers- chromogranin A, pancreatic polypeptide and neuron specific enolase.

The following section covers the specific circulating markers-gastrin, insulin, serotonin, 5-HIAA, glucagon, somatostatin, VIP, ACTH, CRH, GHRH and calcitonin.

The authors develop accordingly on each biomarker and cover current literature.

The modern era of molecular medicine is covered in the following section-advances in molecular biochemical markers. The genomic alteration in NENs and the proteomic biomarkers are reviewed using up-to-date references.

The final section – clinical applicability and outcomes, sums up the data and makes room for the final section-challenges and future directions.

The references used are up-to-date and allow a consistent evaluation of the data concerning the given subject.

Although the paper is not unique in the literature, it summarizes extremely well current data on NET biomarkers with utility in clinical practice.

Author Response

Thank you for your thoughtful and detailed review of our paper on biochemical markers for neuroendocrine tumors. We appreciate your positive feedback on the structure and content, as well as your recognition of the paper’s utility in clinical practice. 

Reviewer 2 Report

Comments and Suggestions for Authors

I-  This is an important review regarding biochemical markers for neuroendocrine tumors. The authors reviewed compared traditional markers and recent development of markers in the field.

2-  The authors did an excellent detailed discussion of the these markers. They discussed the benefits, as well as the some of the problems with some of the traditional markers. But they also described the benefits of new markers, that may be beneficial to neuroendocrine.

3-  This is one of the best reviews that I have seen. Clear, comprehensive, well documented.  Tables 1, 2,  and Figure 1  were also very helpful. 

4-  They also provided a significant up-to-date reference list.

Author Response

Thank you for your positive comments. We are very grateful for your recognition of the effort we put into discussing both traditional and new markers comprehensively.  Your kind words are really appreciated.

Reviewer 3 Report

Comments and Suggestions for Authors

The manuscript entitled "Biochemical markers for neuroendocrine tumors: traditional circulating markers and recent development. A Comprehensive Review" is well-organized and may be published in Diagnostics after some minor changes:

1. I recommend to add a more recent papers to the reference list (last 5 years)

2. The 3rd affiliation is missing at the top of title page

3. I think that 7 self-cited papers of Massironi is to much for 1 paper

4. I recommend to change the title of 3rd Section, because the markers discussed are not specific.

5. 254-256 - adjust, please, the format of text

6. Fig 1 should be corrected (see, miRs block)

Comments on the Quality of English Language

minor

Author Response

Thank you for your constructive feedback, which undoubtedly improved the quality and accuracy of our review.

Here is our point-to-point response:

  1. I recommend to add a more recent papers to the reference list (last 5 years)

We added more recent papers from the last five years to the reference list to ensure the review is current and reflects the latest developments in the field (see ref #88, 105, 115, 132, 133).

  1. The 3rd affiliation is missing at the top of title page

We now added the third affiliation at the top of the title page.

  1. I think that 7 self-cited papers of Massironi is to much for 1 paper

I appreciate the concern regarding the number of self-citations of Massironi's work. We removed citations #1 and 33 (papers by Massironi), and adjusted accordingly to maintain the balance and integrity of the manuscript

  1. I recommend to change the title of 3rd Section, because the markers discussed are not specific.

Your comment on the title of the third section is insightful. Indeed, the substances discussed within this section are not merely markers; they are hormones that are directly responsible for the clinical manifestations of specific neuroendocrine syndromes. These substances are integral to the diagnosis and management of these syndromes, making them "specific" in their application and impact. We have changed the section title from "specific markers" to "Specific Hormones" to more accurately reflect the role these substances play.

  1. 254-256 - adjust, please, the format of text

We adjusted the formatting of the text in the specified lines to meet the journal's standards

  1. Fig 1 should be corrected (see, miRs block)

The issue noted in the miRs block of Figure 1 has been addressed.